# Prognostic factors and quality of life after pelvic fractures. The Brabant Injury Outcome Surveillance (BIOS) study

L. Brouwers[1]*, M. A. C. de Jongh[1,2], L. de Munter[2], M. Edwards[3], K. W. W. Lansink[2,4]

**1** Network Emergency Care Brabant, Elisabeth-Tweesteden Hospital, Tilburg, Noord-Brabant, The Netherlands, **2** Department Trauma Topcare, Elisabeth-Tweesteden Hospital, Tilburg, Noord-Brabant, The Netherlands, **3** Department of Trauma Surgery, Radboud University Medical Center, Nijmegen, Gelderland, The Netherlands, **4** Department of Surgery, Elisabeth-Tweesteden Hospital, Tilburg, Noord-Brabant, The Netherlands

* L.brouwers@etz.nl

## Abstract

### Introduction

Pelvic fractures can have long-term consequences for health-related quality of life (HRQoL). The main purpose of this study is to provide insight into short-term HRQoL in the first year after pelvic injury and to identify short-term prognostic factors of decreased outcome.

### Methods

This is a prospective, observational, multicenter, follow-up cohort study in which HRQoL and functional outcomes were assessed during 12-month follow-up of injured adult patients admitted to 1 of 10 hospitals in the county of Noord-Brabant, the Netherlands. The data were collected by self-reported questionnaires at 1 week (including preinjury assessment) and 1, 3, 6 and 12 months after injury. The EuroQoL-5D (EQ-5D), visual analog scale (VAS), Merle d'Aubigné Hip Score (MAHS) and Majeed Pelvic Score (MPS) were used. Multivariable mixed models were used to examine the course of the HRQoL and the prognostic factors for decreased HRQoL and functional outcomes over time.

### Results

A total of 184 patients with pelvic fractures were identified between September 2015–September 2016; the fractures included 71 Tile A, 44 Tile B and 10 Tile C fractures and 59 acetabular fractures. At the pre-injury, 1 week, and 1, 3, 6 and 12 months after injury time points, the mean EQ-5D Index values were 0.90, 0.26, 0.45, 0.66, 0.77 and 0.80, respectively, and the mean EQ-VAS values were 83, 45, 57, 69, 75 and 75, respectively. At 6 and 12 months after injury, 22 and 25% of the MPS < 65 year group, 38 and 47% of the MPS $\geq$ 65 year group and 34 and 51% of the MAHS group, respectively, reached the maximum score. Pre-injury score, female gender and high Injury Severity Score (ISS) were important prognostic factors for a decreased HRQoL, and the EQ-5D VAS β = 0.43 (95% CI: 0.31 − 0.57), -6.66 (95% CI: -10.90 − -0.43) and -7.09 (95% CI: -6.11 − -5.67), respectively.

**Data Availability Statement:** Data cannot be shared publicly because data from this study can contain potentially identifying or sensitive patient information. Data is anonymized, but due to

relatively few severe cases, patients could be identified. This restriction has been imposed by the Medical Ethics Committee of Brabant. Therefore, data from the BIOS-study will be made available for researchers who meet the criteria for access to confidential data. Requests may be sent to: secretariaat@nazb.nl.

**Funding:** ZonMw (80-84200-98-14255). The funders had no role in study design, data collection and analysis, decision to publish, or preparation of the manuscript. https://www.zonmw.nl/nl/onderzoek-resultaten/kwaliteit-van-zorg/programmas/project-detail/topzorg/prevalence-recovery-patterns-and-risk-factors-of-non-fatal-outcome-and-costs-after-trauma-a-prospe/resultaten/.

**Competing interests:** The authors have declared that no competing interests exist.

## Discussion

Patients with pelvic fractures experience a reduction in their HRQoL. Most patients do not achieve the HRQoL of their pre-injury state within 1 year after trauma. Prognostic factors for decreased HRQoL are a low pre-injury score, high ISS and female gender. We do not recommend using the MAHS and MPS in mid- or long-term follow-up of pelvic fractures because of ceiling effects.

Trial registration number NCT02508675.

## Introduction

Pelvic fracture is a collective name for pelvic ring fractures and acetabular fractures and can occur as a result of both high- and low-energy trauma. In young patients, these injuries normally occur due to a high-energy trauma, [1] whereas in elderly patients, these fractures occur more often due to low-energy trauma. [2,3] In the Netherlands, the current annual incidence of pelvic fractures in the elderly population ($\geq$ 65 years) is nine-fold higher (57.9 per 100.000 inhabitants) than that the younger population. [4] Pelvic fractures have long-term consequences for health-related quality of life (HRQoL) in both younger [5] and elderly patients. [6]

Several studies have reported the health status of trauma patients in a general trauma population and found little improvement beyond 9 to 12 months after minor injury, [7,8] while patients with major injury showed continuous improvement in HRQoL for up to 2 years after injury. [9,10]

In 2010 and 2012, Borg et al. demonstrated a substantially lower HRQoL 2 years after the surgical treatment of pelvic fractures compared with a reference population. [5,11] In 2015, Gabbe et al. showed that 2 years after injury, 77% of patients with severe pelvic ring fractures were living independently, and 59% had returned to work. [12] These authors advised a large-scale multicenter study to fully understand the burden of severe pelvic ring fractures.

However, to the best of our knowledge, no prospectively designed multicenter study has been performed in which HRQoL is investigated during the first year after pelvic injury. A better understanding of the HRQoL and the burden of pelvic injury is crucial to improve the quality of healthcare provided to pelvic trauma patients. [13] To identify prognostic factors for HRQoL after pelvic injury, a longitudinal study with both a generic questionnaire and disease-specific instruments are needed. [13] The main purpose of this study was to gain insight into short-term HRQoL in the first year after pelvic injury. Our other aim was to identify short-term prognostic factors of outcome after pelvic trauma.

## Methods

### The Brabant Trauma Registry

The Dutch Noord-Brabant region has 2.5 million inhabitants. Approximately 12,000 injured trauma patients are admitted and included in the Brabant Trauma Registry (BTR) annually. [14] The BTR includes 12 emergency departments (ED) of Network Emergency Care Brabant, including 1 level 1 trauma center, and compiles prehospital and hospital data of all trauma patients admitted after presentation to the ED.

### BIOS study

This study was approved by the Medical Ethics Committee of Brabant (project number NL50258.028.14). Participants provided their written informed consent to participate in this

study. This pelvic fracture study, in which the HRQoL is assessed during 12 months of follow-up after patient admittance to one of the hospitals, is part of the Brabant Injury Outcome Surveillance (BIOS) study. [15] The BIOS is a prospective longitudinal follow-up study among all admitted adult injury patients ($\geq$ 18 years) in the Noord-Brabant region, regardless of the severity or classification of the injury, to evaluate the total non-fatal burden of injury from the patient and societal perspectives. Patients who had sufficient knowledge of the Dutch language and completed the questionnaires after 1 week, 1 month or 3 months were included in the study. The exclusion criteria were patients with a pathological fracture caused by a malignancy or metastasis or patients older than 80 years. If patients were incapable of completing the self-reported measures themselves because of mental retardation, dementia or other neurological conditions, the questionnaires were completed by a proxy informant. For a detailed description of the study, we refer to the previously published study protocol. [15] The inclusion period in this study was 1 year, from September 1st, 2015, until September 30st, 2016.

## Data collection

Patient characteristics, injury characteristics, additional injuries, complications during admission and possible surgical procedures were extracted from the BTR. We used the Abbreviated Injury Scale (AIS-90, update 2008) [16] to define the anatomical location and severity of the additional injuries. We calculated the Injury Severity Score (ISS) to assess the overall injury severity. [17] The AO/OTA classification was used by the principal investigator to classify pelvic ring fractures into stable fractures (61A1–A3), rotationally unstable fractures (61B1–B3), or rotationally and vertically unstable fractures (61C1–C3) and acetabular fractures into partial articular, isolated column and/or wall fractures (62A), partial articular, transverse type fractures (62B) or complete articular, associated both column fractures (62C). [18]

## Outcome measures

**Generic HRQoL.**  The EQ-5D questionnaire, [19] defined along five dimensions, including mobility, self-care, usual activities, pain or discomfort, and anxiety or depression, with 3 levels each (no, moderate or severe problems) was used to measure the generic HRQoL. The EQ Visual Analogue Scale (EQ VAS) records the patient's self-rated state of health on an analogue scale between 0 (worst imaginable health state) and 100 (best imaginable health state).

A Dutch scoring algorithm (EQ-5D index score) is available by which each health status description can be expressed into a summary index score. [20,21] This index score ranges from -0.329 to 1, in which 0 represents death, 1 represents full health and $< 0$ represents a health state considered worse than death. The EQ-5D index and VAS score of patients with a pelvic fracture and the average EQ-5D index summary score and VAS score for the general Dutch population were compared. [20,21]

**Disease-specific HRQoL.**  Several specific questionnaires have been designed to investigate the HRQoL after pelvic injury. The Majeed Pelvic Score (MPS) is a frequently used pelvic ring-specific HRQoL instrument. [22,23] Recently, the Dutch norm scores for the MPS were collected for the $< 65$ years (88.3) and $\geq$65 years (72.0) age groups. [24] The Merle d'Aubigné Hip Score (MAHS) is a commonly used questionnaire to evaluate functional results after acetabular fractures. [25–27] We used the MPS to measure HRQoL in patients with pelvic ring fractures and the MAHS to measure HRQoL in patients with isolated acetabular fractures.

**MPS.**  The MPS questionnaire [22] is defined along seven dimensions, including pain, work, sitting, sexual intercourse and standing, (walking aids, unaided gait, walking distance), with scores of 5/30, 0/20, 4/10, 1/4 and 6/36, respectively (2/12, 2/12, 2/12)(minimum/maximum points). The MPS ranges between 16 (worst health state) and 100 (best health state). The

patients were divided into <65 years (working) and ≥65 years (retired) age groups, with maxima of 100- and 80 points, respectively. [28]

**MAHS.** The MAHS is a clinical hip score that evaluates pain, ambulation and mobility. The pain and ambulation domains are divided into 6 grades, where 1 indicates the worst and 6 indicates the best state of the patient. [25] The domain mobility/range of motion (ROM) is determined by comparison of the total score for the injured side with that for the uninjured side (flexion, abduction, adduction and rotation). This domain is divided into 5 grades (0–39, 40–59, 60–79, 80–94, and 95–100% ROM), with points given as 1, 3, 4, 5 and 6, respectively. The total minimum score is 3, and the maximum is 18; Excellent is indicated by 18, Good by 15–17, Fair by 12–14, and Poor by 3–11.

## Follow-up

Patients received the information letter, informed consent form, pre-injury questionnaire and first questionnaire within the first week of hospital stay or at their home address. Patients could choose between returning the questionnaires online or with paper and pencil. The EQ-5D data were collected at 1 week and 1, 3, 6 and 12 months after trauma. The MPS data were collected 1, 3, 6 and 12 months after trauma. The MAHS data were collected at 6 weeks and 3, 6 and 12 months after trauma. Patients were considered lost to follow-up if the questionnaires were not completed from any follow-up time point permanently. The level of education was included in the questionnaire according to the Dutch standards of Statistics Netherlands (Centraal Bureau voor Statistiek): low education level (highest degree basisonderwijs of vmbo, mbo 1, havo onderbouw), intermediate education level (havo, vwo, mbo 2, 3, 4) or a high education level (hbo, wo bachelor, wo master, doctor).

## Data analysis

Descriptive statistics were calculated to provide an overview of the characteristics of the study population. The maximum scores of the MPS and MAHS were assessed. Ceiling effects were considered to be present if >15% of respondents achieved the highest possible score. [23,29] Multivariable mixed models were used to examine the course of HRQoL and prognostic factors for decreased HRQoL and functional outcomes over time. Patient characteristics, self-reported pre-injury HRQoL and injury-related characteristics, i.e., ISS, pelvic operation (yes or no) and low-energy trauma (LET) or high-energy trauma (HET), were tested as prognostic factors of decreased HRQoL. The ISS was categorized into the 1–8, 9–15 and >15 groups. Age was categorized into the <65 (N = 97) and ≥ 65 years (N = 87) groups. Statistical test results were considered significant at a level of p<0.05. All analyses were conducted using SPSS V.24.0 (Statistical Package for Social Sciences, Chicago, Illinois, USA).

## Results

### Inclusion and exclusion

A total of 204 patients with pelvic fractures were admitted (Fig 1). Ninety percent of the patients (N = 184) were included in the study. Twelve patients did not want to participate. Eight patients were excluded; 1 patient was lost to follow-up soon after discharge from the hospital, and 7 patients were excluded because of a pre-injury poor mental state without the availability of a proxy informant.

Almost all the included patients completed their follow-up. However, during follow-up, 4 patients died as a result of cardiovascular of cardiorespiratory diseases and their advanced age, and 1 died due to suicide. Eight patients showed no interest in participating anymore during

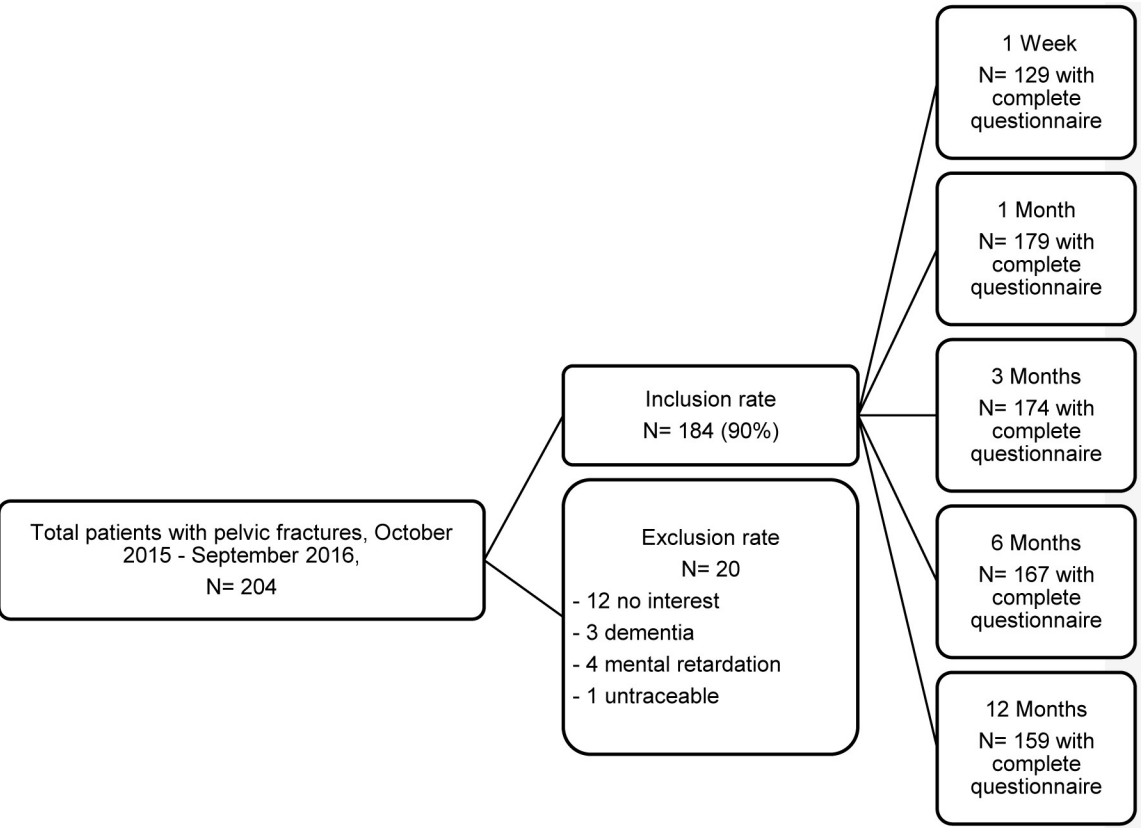

**Fig 1. Flow of participants through the study.**

follow-up. All data received for patients who did not complete their follow-up were used during our analysis.

## Patient characteristics

Mean age of females in our study was 62 (SD 16) years and mean age for male patients was 56 (SD 17) years (P = 0.03). Mean pre-injury EQ5D index score of female patients was 0.85 (SD 0.20) while the score of male patients was 0.93 (SD 0.16), a significant difference (P = 0.01).

The patient characteristics are shown in Table 1. The pelvic fractures were divided over the two AO/OTA groups (61 and 62 respectively). Eighty-seven percent (N = 62) of the 61A fractures were minimally displaced fractures of the ring (62A2). Eighty percent (N = 35) of the 61B fractures were identified as lateral compression injuries (61B2), and 70% (N = 7) of the 61C fractures were unilateral, rotationally and vertically unstable (61C1). Eleven acetabular fractures were classified as posterior wall types, 2 as anterior wall, 6 as anterior column, 6 as transverse, 14 as T-type, 2 as posterior wall/posterior column, 6 as anterior column/posterior hemitransverse and 12 as both column. This resulted in 21 62A, 26 62B and 12 62C type fractures according to the AO/OTA classification.

Eighty-two patients (45%) had a low education level, 62 (34%) an intermediate education level and 37 (20%) a high education level. In the general Dutch working population, 29% have a low education level, 40% have an intermediate education level and 30% have a high education level. [30]

**Table 1. Patient characteristics.**

| | 61A | 61B | 61C | Acetabulum |
|---|---|---|---|---|
| N (%) | 71 (39) | 44 (24) | 10 (5) | 59 (32) |
| Pelvic AO/OTA subtypes N (%) | | | | |
| .1 | 0 (0) | 1 (2) | 7 (70) | |
| .2 | 62 (87) | 35 (80) | 2 (20) | |
| .3 | 9 (13) | 8 (18) | 1 (10) | |
| Acetabulum AO/OTA subtypes N (%) | | | | |
| 62A | | | | 21 (36) |
| 62B | | | | 26 (44) |
| 62C | | | | 12 (20) |
| Gender | | | | |
| % male | 41 | 57 | 60 | 81 |
| Age | | | | |
| Mean Years (SD) | 63 (16) | 53 (19) | 44 (19) | 59 (14) |
| Mechanism in % | | | | |
| Fall from same level | 63 | 27 | 0 | 32 |
| Fall from height | 17 | 25 | 50 | 24 |
| Traffic accident | 18 | 43 | 40 | 41 |
| Entrapment | 2 | 5 | 10 | 3 |
| Trauma mechanism in % | | | | |
| Low Energy Trauma | 62 | 23 | 0 | 32 |
| High Energy Trauma | 38 | 77 | 100 | 68 |
| Associated injuries (AIS severity>1) % | | | | |
| AIS region Head | 14 | 23 | 0 | 19 |
| AIS region Face | 11 | 9 | 10 | 3 |
| AIS region Neck | 0 | 0 | 0 | 0 |
| AIS region Thorax | 11 | 25 | 40 | 17 |
| AIS region Abdomen | 4 | 16 | 40 | 7 |
| AIS region Spine | 13 | 18 | 70 | 5 |
| AIS region Upper extremity | 25 | 27 | 20 | 29 |
| AIS region Lower extremity | 100 | 100 | 100 | 100 |
| AIS region Unspecified | 7 | 7 | 30 | 29 |
| Shock type %* | | | | |
| Type 1 | 96 | 61 | 10 | 80 |
| Type 2 | 4 | 32 | 60 | 18 |
| Type 3 | 0 | 7 | 20 | 2 |
| Type 4 | 0 | 0 | 10 | 0 |
| Mean ISS (SD) | 7 (6) | 13 (12) | 26 (12) | 8 (6) |
| Pelvic operation % | 1 | 39 | 100 | 42 |
| Median hospitalization in days (IQR) | 5 (6) | 9 (9) | 25 (28) | 10 (10) |
| Neurological complication % | 1 | 16 | 30 | 7 |
| Total hip arthroplasty < 1 year % | 0 | 0 | 0 | 12 |
| 1-year mortality, N (%) | 2 (3) | 0 | 1 (10) | 2 (3) |

*Types of hemorrhagic shock according to ATLS guidelines.

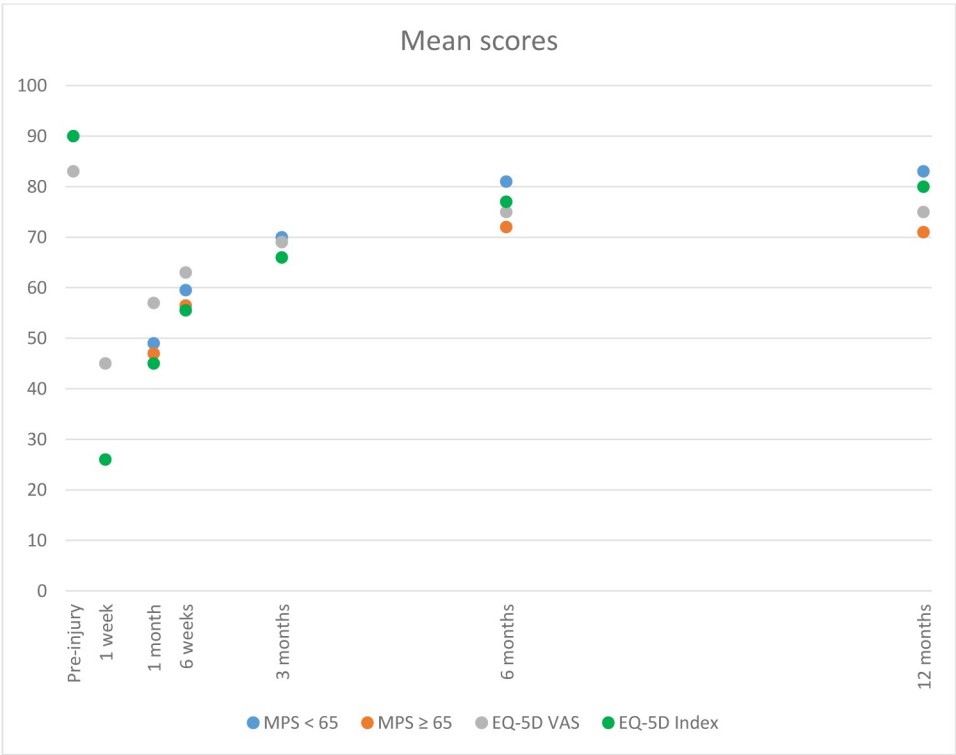

**Fig 2. Mean EQ-5D VAS, index score\* and MPS over time.** \*To combine all questionnaires in one figure, the results of the EQ-5D index are multiplied by a hundred ("x 100").

## Mean questionnaire scores

Fig 2 shows a global graphic of the mean EQ-5D VAS, index and MPS at each time point post-injury. Pre-injury EQ-5D index and VAS scores were 0.90 and 83, respectively. The average EQ-5D index summary and VAS scores for the general Dutch population were, respectively, 0.87, SD 0.18 and 77.72, SD 15.19.(30,31) At 1 week post-injury, patients scored means of 0.26 and 45, respectively. Patients scored means of 0.45, 57, 49 and 47 points on the Index, VAS and < 65 and ≥ 65 MPS, respectively, at one month after injury. At 3, 6 and 12 months after injury, patients scored 0.66/69/70/66, 0.77/75/81/72 and 0.80/75/83/70 points, respectively.

Fig 3 shows the mean values. At 6 weeks and 3, 6 and 12 months after injury, patients with an acetabular fracture scored means of 11.38, 14.02, 15.75 and 16.79 points on the MAHS, respectively.

## Maximum MPS and MAHS

In general, very few maximum scores of the MPS were seen at 1 month after injury in both age groups (Table 2). Three months after injury, 9% of the patients < 65 years scored a maximum score of 100 points, while 23% of the patients ≥65 years scored the maximum score of 80 points. At 12 months after injury, almost 50% of the patients ≥65 years had a maximum score on the MPS. At 6 and 12 months after trauma 34% and 50% of patients with an acetabular fracture had a maximum MAHS, respectively (Table 3).

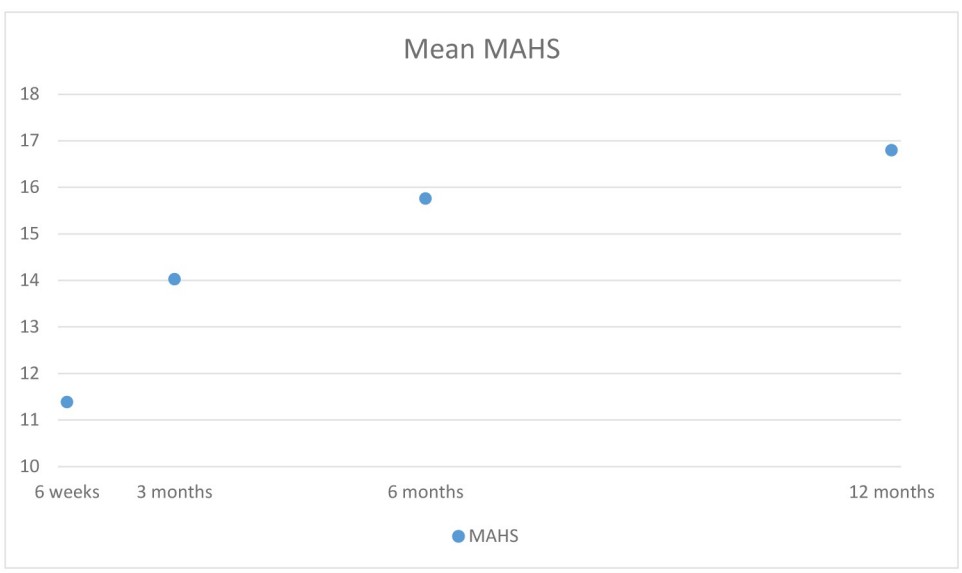

**Fig 3. Mean MAHS over time, total mean score.**

**Table 2. Mean MPS (standard deviation) scores and frequency of patients with maximum scores.**

| Time point | Mean (SD) < 65 years | Mean (SD) ≥ 65 years | Maximum score (%) < 65 years | maximum score (%) ≥ 65 years |
|---|---|---|---|---|
| 1 month | 49 (16) | 47 (13) | 0 (0) | 1 (2) |
| 3 months | 70 (19) | 66 (13) | 5 (9) | 10 (23) |
| 6 months | 81 (17) | 72 (10) | 12 (22) | 15 (38) |
| 12 months | 83 (17) | 70 (13) | 13 (25) | 21 (47) |

## HRQoL outcomes over time and prognostic factors of decreased HRQoL

Tables 4, 5 and 6 show regression coefficients (95% CI) from the multivariable mixed models investigating the predictors of reporting problems on the EQ-5D index, VAS, MPS and MAHS, respectively. The outcomes of the EQ-5D index and VAS questionnaires 1, 3, 6 and 12 months after trauma were compared to the outcome 1 week after trauma.

The pre-injury score was an important prognostic factor for a decreased HRQoL in the EQ-5D VAS and index score. Female gender was associated with a lower score on the EQ-5D VAS (β = -6.67 (95% CI: -10.90 − -0.43)) compared with male gender. Patients with a higher ISS were associated with a lower score on the EQ-5D VAS and Index score (respectively β = -7.10 (95% CI: -12.41 − -1.78) and β = -0.11 (95% CI: -0.19 − -0.03)) compared with those with a lower ISS.

**Table 3. Mean MAHS (standard deviation) scores and frequency of patients with maximum scores.**

| Time point | Mean (SD) | maximum score (%) |
|---|---|---|
| 6 weeks | 11.4 (2.9) | 0 (0) |
| 3 months | 14.0 (2.5) | 5 (9) |
| 6 months | 15.8 (2.6) | 20 (34) |
| 12 months | 16.8 (1.8) | 30 (51) |

**Table 4. The HRQoL outcome over time and prognostic factors in the first year.** Regression coefficients (95% CI) for EQ-5D and VAS for the first year after trauma assessed with multivariable mixed models.

| | EQ-5D index score[#] | | | EQ-5D VAS score[#] | | |
|---|---|---|---|---|---|---|
| | Beta | 95% CI | p-value | Beta | 95% CI | p-value |
| **1 week** | 0[*] | - | - | 0[*] | - | - |
| **1 month** | 0.21 | 0.16 – 0.25 | < 0.001 | 14.23 | 10.95 – 17.51 | < 0.001 |
| **3 months** | 0.43 | 0.38 – 0.48 | < 0.001 | 26.67 | 23.38 – 29.95 | < 0.001 |
| **6 months** | 0.53 | 0.49 – 0.58 | < 0.001 | 32.21 | 28.86 – 35.56 | < 0.001 |
| **12 months** | 0.56 | 0.52 – 0.61 | < 0.001 | 32.80 | 29.44 – 36.17 | < 0.001 |
| **Female** | -0.06 | -0.13 – 0.00 | 0.05 | -6.67 | -10.90 – -2.43 | < 0.001 |
| **HET** | -0.03 | -0.10 – 0.05 | 0.52 | -1.81 | -6.86 – 3.23 | 0.48 |
| **Pelvic operation** | -0.02 | -0.09 – 0.05 | 0.53 | -2.22 | -6.93 – 2.49 | 0.35 |
| **Age ≥ 65 years** | 0.017 | -0.05 – 0.08 | 0.61 | 2.83 | -1.52 – 7.19 | 0.20 |
| **ISS 1–8** | 0[*] | - | - | 0[*] | - | - |
| **ISS 9–15** | -0.03 | -0.12 – 0.06 | 0.46 | -0.22 | -6.12 – 5.67 | 0.94 |
| **ISS ≥ 16** | -0.11 | -0.19 – -0.03 | < 0.01 | -7.10 | -12.41 – -1.78 | < 0.01 |
| **Low education level** | 0[*] | - | - | 0[*] | - | - |
| **Intermediate education level** | 0.05 | -0.02 – 0.11 | 0.19 | 0.80 | -3.69 – 5.29 | 0.73 |
| **High education level** | 0.03 | -0.05 – 0.11 | 0.43 | -1.39 | -6.60 – 3.82 | 0.60 |
| **Pre-injury score[a]** | 0.61 | 0.43 – 0.79 | < 0.001 | 0.44 | 0.31 – 0.57 | < 0.001 |

[*] reference group.

[a] pre-injury EQ-5D index score and pre-injury EQ-5D VAS score for respectively EQ-5D index score and EQ-5D VAS score

[#] adjusted for all other variables in the table

Beta = regression coefficient, ISS = Injury Severity Score, CI = Confidence Interval, HET = High Energy Trauma

Females ≥ 65 years with pelvic ring fractures were associated with a lower MPS (β = -7.79 (95% CI: -14.44 − -1.14)) compared with males. Patients ≥ 65 years with a pelvic ring fracture with a higher ISS were associated with lower scores on the MPS β = -13.76 (95% CI: -24.27 − -3.24)) compared with those with a lower ISS. Pelvic type, acetabulum type, pelvic operation, high- or low-energy trauma and education level were not significantly associated with lower or higher HRQoL after pelvic injury.

## Discussion

This study was performed to gain more insight into short- and mid-term HRQoL after pelvic injury. Furthermore, prognostic factors of decreased outcome in the first year after pelvic injury were identified. Patients with pelvic fractures experience a severe reduction in their HRQoL and functional outcomes, especially within the first 3 months after injury. Although patients recover up to 12 months after trauma, most patients do not reach their pre-injury status. Prognostic factors for a decreased quality of life after pelvic trauma are high ISS, low pre-injury HRQoL status and female gender.

### Short- and mid-term HRQoL

The EQ5D-index and VAS outcomes of pelvic fracture patients in our study showed steep decreases in the first week and steep increases in the first 3 months after trauma when compared with their pre-injury status. Although the HRQoL recovery continues to improve up to 12 months after trauma, most patients do not achieve their pre-injury state of HRQoL. One

**Table 5. The HRQoL outcome over time and prognostic factors in the first year, separate for patients younger than 65 years and 65 years or older, with multivariable mixed models.**

| | MPS < 65 years[a] | | | MPS ≥ 65 years[a] | | |
|---|---|---|---|---|---|---|
| | Beta | 95% CI | p-value | Beta | 95% CI | p-value |
| **61A** | 0* | - | - | 0* | - | - |
| **61B** | -1.92 | -12.19 − 8.35 | 0.71 | -0.65 | -7.89 − 6.59 | 0.86 |
| **61C** | -12.23 | -31.63 − 7.18 | 0.21 | 9.91 | -8.47 − 28.28 | 0.28 |
| **1 month** | 0* | - | - | 0* | - | - |
| **3 months** | 19.49 | 15.06 − 23.91 | < 0.001 | 18.64 | 14.91 − 22.37 | < 0.01 |
| **6 months** | 31.52 | 27.13 − 35.92 | < 0.001 | 23.50 | 19.71 − 27.28 | < 0.01 |
| **12 months** | 33.77 | 29.32 − 38.22 | < 0.001 | 23.45 | 19.77 − 27.14 | < 0.01 |
| **Female** | -7.44 | -16.47 − 1.59 | 0.10 | -7.79 | -14.44 − -1.14 | 0.02 |
| **HET** | 2.09 | -9.60 − 13.78 | 0.72 | 3.47 | -4.50 − 11.44 | 0.38 |
| **Pelvic operation** | 4.79 | -6.69 − 16.27 | 0.41 | -6.63 | -18.96 − 5.70 | 0.28 |
| **ISS 1–8** | 0* | - | - | 0* | - | - |
| **ISS 9–15** | 5.98 | -6.10 − 18.06 | 0.33 | -3.53 | -12.21 − 5.14 | 0.42 |
| **ISS ≥ 16** | -5.99 | -16.78 − 4.80 | 0.27 | -13.76 | -24.27 − -3.24 | 0.01 |
| **Low education level** | 0* | - | - | 0* | - | - |
| **Intermediate education level** | 5.12 | -4.00 − 14.24 | 0.27 | 2.69 | -3.85 − 9.22 | 0.41 |
| **High education level** | 7.89 | -3.04 − 18.82 | 0.15 | 2.84 | -4.94 − 10.62 | 0.47 |

*reference group.

[a]adjusted for all other variables in the table

Beta = regression coefficient, ISS = Injury Severity Score, CI = Confidence Interval, HET = High Energy Trauma

**Table 6. The HRQoL outcome over time and prognostic factors in the first year, with multivariable mixed models.**

| | Merle d'Aubigne hip score | | |
|---|---|---|---|
| | Beta | 95% CI | p-value |
| **Elementary fracture** | 0* | - | - |
| **Associated fracture** | -1.16 | -2.36 − 0.03 | 0.06 |
| **6 weeks** | 0* | - | - |
| **3 months** | 2.66 | 2.03 − 3.28 | < 0.001 |
| **6 months** | 4.40 | 3.78 − 5.02 | < 0.001 |
| **12 months** | 5.24 | 4.59 − 5.89 | < 0.001 |
| **Female** | 0.14 | -1.37 − 1.64 | 0.86 |
| **HET** | 1.24 | -0.20 − 2.67 | 0.09 |
| **Pelvic operation** | -0.28 | -1.62 − 1.07 | 0.68 |
| **Age ≥ 65 years** | -0.04 | -1.33 − 1.26 | 0.96 |
| **ISS 1–8** | 0* | - | - |
| **ISS 9–15** | -0.12 | -2.28 − 2.04 | 0.91 |
| **ISS ≥ 16** | -0.41 | -2.14 − 1.32 | 0.63 |
| **Low education level** | 0* | - | - |
| **Intermediate education level** | -0.01 | -1.53 − 1.52 | 0.99 |
| **High education level** | 0.78 | -0.98 − 2.54 | 0.38 |

*reference group.

Beta = regression coefficient, ISS = Injury Severity Score, CI = Confidence Interval, HET = High Energy Trauma

year of follow-up could be insufficient to reach a pre-injury status. This hypothesis could be confirmed by the index score of the general Dutch population, which is 0.87, [20] while our mean index score was 0.80 at 12 the month time point after trauma.

Borg et al. observed a substantially lower HRQoL in patients with surgically treated 61B and C fractures two years after injury compared with a reference population. [5] Giannoudis et al. observed a mean EQ-5D index score of 0.73 (population norm score 0.85) and a mean VAS score of 71.5 in patients with operatively treated isolated acetabular fractures with a mean follow-up of 36 months. [31] Although these studies were not completely comparable to our study (only 29% surgically treated patients), they showed a decreased HRQoL of at least one year after trauma and possibly up to even 2 years after trauma.

For the MPS and MAHS, three months after trauma seems to be a turning point. The recovery curves of the MPS ≥ 65 years and MAHS plateau after this time point, while the recovery curves of the MPS < 65 years patients only plateau 6 months after trauma. This observation could mean that younger patients need more time to recover, probably due to more severe trauma. The older group seems to "stagnate" at 3 months after trauma in terms of recovery, although this stagnation could mean that this group is almost fully recovered at that time due to their less severe injury. This hypothesis of the relationship between age and trauma severity could be confirmed by the patient characteristics: older patients are more frequently seen in the 61A group (low-energy trauma) compared with the 61B and C group (high-energy trauma).

## Prognostic factors that could influence HRQoL after pelvic trauma

Prognostic factors that are known to influence the quality of life after pelvic trauma include neurological impairment of the lower extremities, aging, complex fracture type, surgery, chronic pain and sexual and urological dysfunction. [32–35] Patients with isolated acetabular fractures are at risk for a decreased HRQoL especially due to osteoarthritis, heterotopic ossification and avascular necrosis of the femoral head. [36] However, most studies that focus on prognostic factors after a pelvic trauma are single-center, retrospective, and cross-sectional in nature or feature small sample sizes with a follow-up starting 1 year after trauma. [23,28,29,34,35,37,38]

Our study shows several prognostic factors that lead to decreased HRQoL after pelvic trauma. A low pre-injury HRQoL status seems to be an important prognostic factor for both acetabular- and pelvic ring fractures. However, the timing of the pre-injury score has been debated in earlier studies. Williamson et al. [39] concluded that it was allowed to implement a pre-injury score up to 6 months after trauma. Hernefalk et al. concluded that completing the pre-injury HRQoL questionnaire 1–2 months after trauma was more accurate and that pre-injury assessments were possibly susceptible to distortion. [40] The pre-injury VAS scores of patients with surgically treated acetabular and pelvic ring fractures were calculated by these authors to be 79, 85 and 86 at 1 week, 1 and 2 months after injury, respectively. In our study, we found a mean pre-injury VAS score of 83, which is comparable with the results of Hernefalk 1–2 months after injury. Williamson et al. demonstrated that patients with increasing age (> 65 years) reported a higher pre-injury status at 12 months post-injury when compared with the pre-injury status reported earlier in the year after trauma. Thus, although the studies of Hernefalk et al., Williamson et al. and our study showed that the timing of the pre-injury questionnaires is arbitrary, our study demonstrated that the pre-injury assessment is important to measure and that a low pre-injury status is a risk factor for a decreased HRQoL for young and old patients.

Gender is an also important prognostic factor. A significant difference was found in the EQ5D VAS score, and specific for the older pelvic ring fracture patients (MPS ≥ 65). No

significant difference was found in the subset of acetabular patients (MAHS). This could imply that women of $\geq$ 65 years with a pelvic ring fracture are more prone for a worse HRQoL.

In contrast to the study of Holstein et al., [34] we found the female gender to be a prognostic factor for a reduced quality of life after sustaining a pelvic fracture. Polinder et al. and Holbrook et al. [8,41] found female gender to be a prognostic factor for a poor HRQoL after trauma in general. Holbrook concluded that a better understanding of the impact of major trauma in men and women will be an important component of efforts to improve trauma care and long-term outcome in mature trauma systems. It could be that females in general have a lower pre-injury score when compared with male.

To gain a better understanding of the HRQoL and burden of pelvic injury, it is important to investigate the overall impact of the injury and to determine the ISS. In our study, we measured ISS, including the AIS of the pelvic fracture, meaning that the ISS could be slightly influenced by the gradation of the pelvic fracture. However, it is known that patients with pelvic fractures suffer from many associated injuries. [42] We found ISS to be a prognostic factor for a decreased HRQoL.

Literature has shown that aging could be a prognostic factor for a reduced quality of life after pelvic trauma [34] and trauma in general. [13] However, except for females in the MPS $\geq$ 65 group, we did not find a relationship between aging and a decreased HRQoL when using the EQ5D Vas, Index score or MAHS. An important reason for this lack of association in the combined pelvic ring/ acetabular fracture group could be the inclusion of a pre-injury HRQoL questionnaire in our longitudinal analysis, which was not included in the questionnaires of other studies. Our hypothesis is that elderly patients with multiple comorbidities will score lower on the pre-injury questionnaire compared with the healthy young population. Therefore, the pre-injury score is a stronger prognostic factor for a reduced HRQoL than age.

Pelvic fracture type, acetabular fracture type, HET or LET, education level and pelvic operation were not prognostic factors. It is possible that the sample size of patients was insufficient to draw conclusions about these prognostic factors. It could also be possible that prognostic factors such as pelvic or acetabular type and operation are of importance to the follow-up during the first months after trauma. Prognostic factors were assessed over the first year after injury; prognostic factors for short-term recovery could be leveled out if they were not also prognostic factors for long-term recovery. Therefore, a new study is needed with larger sample sizes to draw conclusions using these prognostic factors.

### Strengths and limitations

The strengths of our study include the multicenter prospective longitudinal design, high response rate and low prevalence of missing data. To the best of our knowledge, this is the largest longitudinal study of the follow-up of patients with pelvic fractures. A longitudinal design has several benefits [43]: 1). following the change in individual patients over time, 2). recording the sequence of an event, 3). avoiding recall bias by its prospective nature and 4). relating exposures to event. A disadvantage of this study design might be an incomplete or loss to follow-up of individuals. However, in our study, both the inclusion rate and follow-up rate were high. We are aware of the fact we earlier concluded that Tile C patients <65 years had significantly lower EQ-5D index and total MPS scores.(28) We now feel that a longitudinal analysis is much more comprehensive in investigating HRQoL patterns when compared with just one follow-up moment without a pre-injury score.

Our study also has important limitations. Few patients with 61C fractures were included. Therefore, we were not able to perform a sub-analysis of the AO/OTA-61 and 62 groups. Furthermore, other possible prognostic factors were not collected (i.e., quality of surgical fracture

reduction). Other studies have debated the importance of the relationship between surgical fracture reduction and HRQoL outcomes. [44–46] Prognostic factors were assessed over the first year after injury; prognostic factors for short-term recovery could be leveled out if they were not also prognostic factors for long-term recovery. Therefore, a new study is needed with larger sample sizes to draw conclusions using these prognostic factors.

Serious remarks can be made about the disease-specific HRQoL instruments, MPS and MAHS. Although both questionnaires are often used in pelvic or acetabular-related research, they have not been validated by a formal validation process, nor have they been officially translated into Dutch. [47,48] While the pre-injury scores of the EQ5D-VAS and index showed non-completed recovery 12 months after trauma, ceiling effects of the MPS and MAHS were already present at 3 and 6 months after trauma, meaning that neither questionnaire is specific enough to differentiate between specific recovery levels during the follow-up of pelvic fractures and that long-term outcomes are biased. Lefaivre et al. compared with the MPS with another major generic HRQoL instrument, the Short Form-36; these authors also found ceiling effects and questioned the reliability and responsiveness of this approach over time. [47]

Therefore, we do not recommend the use of the MAHS and MPS in the mid- and long-term follow-up of pelvic fractures. More research is needed to develop disease-specific HRQoL questionnaires that are suitable for long-term follow-up.

## Conclusion

Patients with pelvic fractures experience a reduction of their HRQoL, especially in the first 3 months. The HRQoL recovery continues to improve up to 12 months after trauma, and most patients do not achieve their pre-injury state of HRQoL. Prognostic factors for decreased HRQoL after pelvic trauma are a low pre-injury score, high ISS and female gender. A longer follow-up is needed to examine the HRQoL of pelvic fracture patients. We do not recommend the use of MAHS and MPS in the mid- and long-term follow-up of pelvic fractures.

## Supporting information

**S1 Protocol.**
(PDF)

## Acknowledgments

Members of the BIOS group: P.V. van Eerten, F.C. van Eijck, H.J.A.A. van Geffen, W.A.J.J.M. Haagh, L.M.S.J. Poelhekke, J.B. Sintenie, C.T. Stevens, A.H. van der Veen, C.H. van der Vlies and D.I. Vos.

## Author Contributions

**Conceptualization:** L. Brouwers, M. A. C. de Jongh, K. W. W. Lansink.

**Data curation:** M. A. C. de Jongh, K. W. W. Lansink.

**Formal analysis:** L. Brouwers, M. A. C. de Jongh, K. W. W. Lansink.

**Funding acquisition:** K. W. W. Lansink.

**Investigation:** L. Brouwers, M. A. C. de Jongh, L. de Munter, K. W. W. Lansink.

**Methodology:** L. Brouwers, L. de Munter, K. W. W. Lansink.

**Project administration:** L. Brouwers, K. W. W. Lansink.

**Supervision:** M. A. C. de Jongh, M. Edwards, K. W. W. Lansink.

**Validation:** M. A. C. de Jongh.

**Visualization:** M. A. C. de Jongh.

**Writing – original draft:** L. Brouwers.

**Writing – review & editing:** L. Brouwers, M. A. C. de Jongh, L. de Munter, M. Edwards, K. W. W. Lansink.

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
