## [Decision Letter · Decision Letter 0]

23 Mar 2020

PONE-D-20-05544

Prognostic factors and quality of life after pelvic fractures. The Brabant Injury Outcome Surveillance (BIOS) study.

PLOS ONE

Dear MD. MSc brouwers,

Thank you for submitting your manuscript to PLOS ONE. After careful consideration, we feel that it has merit but does not fully meet PLOS ONE’s publication criteria as it currently stands. Therefore, we invite you to submit a revised version of the manuscript that addresses the points raised during the review process.

We would appreciate receiving your revised manuscript by May 07 2020 11:59PM. To enhance the reproducibility of your results, we recommend that if applicable you deposit your laboratory protocols in protocols.io, where a protocol can be assigned its own identifier (DOI) such that it can be cited independently in the future. For instructions see: http://journals.plos.org/plosone/s/submission-guidelines#loc-laboratory-protocols

We look forward to receiving your revised manuscript.

Kind regards,

Zsolt J. Balogh, MD, PhD, FRACS

Academic Editor

PLOS ONE

Journal Requirements:

a) Did participants provide their written or verbal informed consent to participate in this study?

3. Please expand the acronym “ZonMw ” (as indicated in your financial disclosure) so that it states the name of your funders in full.

Reviewers' comments:

Reviewer's Responses to Questions

**Comments to the Author**

1. Is the manuscript technically sound, and do the data support the conclusions?

Reviewer #1: Yes

Reviewer #2: Yes

2. Has the statistical analysis been performed appropriately and rigorously? 

Reviewer #1: Yes

Reviewer #2: Yes

3. Have the authors made all data underlying the findings in their manuscript fully available?

Reviewer #1: Yes

Reviewer #2: Yes

4. Is the manuscript presented in an intelligible fashion and written in standard English?

Reviewer #1: Yes

Reviewer #2: Yes

5. Review Comments to the Author

Reviewer #1: Comments on Prognostic factors and quality of life after pelvic fractures. The Brabant Injury Outcome Surveillance (BIOS) study by Brouwers et al.

The authors performed a study to gain more insight into short- and mid-term HRQoL after pelvic injury. Additionally, prognostic factors of decreased outcome in the first year after pelvic injury were identified. The study was well designed and executed although the overall findings were not very surprising; The authors concluded that patients with pelvic fractures experienced a severe reduction in their HRQoL and functional outcomes with most patients not reaching their pre-injury status after 12 months. Prognostic factors for a decreased quality of life after pelvic trauma were high ISS, low pre-injury HRQoL status and female gender.

I have a few questions:

• Why did you include patients with acetabular fractures? As you already pointed out in the discussion patients with acetabular fractures have different issues that can influence HRQoL like osteoarthritis, heterotopic ossification and avascular necrosis of the femoral head. Would you consider a sub analysis of patients with acetabular fractures? If not, why not?

• How do you explain that type pelvic type, acetabulum type, pelvic operation, high- or low-energy trauma were not significantly associated with lower or higher HRQoL after pelvic injury. It is difficult to understand that a patient with for example a type C pelvic fracture who underwent extensive surgery would have a similar HRQoL compared with a patient with conservatively treated type A pelvic fracture. In a paper published in Injury 2018 (ref 28) you conclude that Tile C patients <65 years had significantly lower EQ-5D index and total MPS scores. Please explain this inconsistency.

• You demonstrated that female gender was associated with a lower HRQoL, but age was not. In the discussion you state “we did not find a relationship between aging and a decreased HRQoL. An important reason for this lack of association could be the inclusion of a preinjury HRQoL questionnaire in our longitudinal analysis, which was not included in the questionnaires of other studies”. Since elderly patients were more likely female, how do you explain that age was not a prognostic factor. Is it possible that females had lower preinjury HRQoL? Please explain

Other remarks:

In the methods section on page 4: September has only 30 days

.

Reviewer #2: Thank you for the opportunity to review this paper.

This is a prospective multicentre cohort observational study examining pelvic fracture impact on HRQoL for 12 months post injury utilising a series of scoring systems. The prospectively maintained database followed post-injury trauma patients with generic and injury specific QOLs. Th authors conclude that low pre-injury scores, female gender and higher ISS are prognostic for lower post-injury scores.

The authors should be commended on their work. There are a few areas that would need to be addressed before publication.

The exclusion criteria includes patients over 80. This is a very common age group to sustain low energy pelvic/acetabular injuries. What were the reasons for exclusion of this particular group?

The impact of concomitant injuries and their impact upon their outcome scores could use more exploration. Whilst stratified for ISS there seems little discussion on the obvious effect that other injuries would have upon patient outcomes.

The use of the retrospective pre-injury score is fraught with some difficulties as noted by the authors. It is noted to be a prognostic factor for low post-injury scores. It is however unclear if this is a globally reduced score, or whether it is a reduced score relative to their baseline. It is unclear whether this is calculated relative to their pre-injury scores or whether this is calculated on the mean (we would not expect their pelvic injury to increase their QoL scores!)

Is there any 24 month data? Its noted in the other BIOS article referenced ( Prevalence, recovery patterns and predictors of quality of life and costs after non-fatal injury: the Brabant Injury Outcome Surveillance (BIOS) study) That 24 month data points were collected. The authors have referenced 24 and 36 month outcomes from other papers, it would useful to include this data for comparison with the other studies referenced in the introduction.

Figure 3, only demonstrates a single data point? Should there be separate points for both <65 and >65? Visually a line graph connecting the data points would be easier to follow than dots.

As noted by the authors, pelvic fractures have a bimodality of injury presentation. Breakdown by high energy and low energy would be useful rather than purely fracture pattern type to highlight the two different disease patterns,

Similarly this would give more meaning to the presence/absence of concomitant injuries and the types of shock on presentation, absence and presence of surgery. Furthermore, indications of requirements for other non-pelvic surgery related to their trauma would be useful, especially in the conservatively managed multi-trauma pelvic fracture patient.

6. PLOS authors have the option to publish the peer review history of their article (what does this mean?). If published, this will include your full peer review and any attached files.

Reviewer #1: No

Reviewer #2: No

---

## [Author Response · Author response to Decision Letter 0]

7 May 2020

I attached a word document with all answers and tables .

---

## [Decision Letter · Decision Letter 1]

12 May 2020

Prognostic factors and quality of life after pelvic fractures. The Brabant Injury Outcome Surveillance (BIOS) study.

PONE-D-20-05544R1

Dear Dr. brouwers,

We are pleased to inform you that your manuscript has been judged scientifically suitable for publication and will be formally accepted for publication once it complies with all outstanding technical requirements.

With kind regards,

Zsolt J. Balogh, MD, PhD, FRACS

Academic Editor

PLOS ONE

Additional Editor Comments (optional):

Reviewers' comments:

Reviewer's Responses to Questions

**Comments to the Author**

1. If the authors have adequately addressed your comments raised in a previous round of review and you feel that this manuscript is now acceptable for publication, you may indicate that here to bypass the “Comments to the Author” section, enter your conflict of interest statement in the “Confidential to Editor” section, and submit your "Accept" recommendation.

Reviewer #1: All comments have been addressed

Reviewer #2: All comments have been addressed

2. Is the manuscript technically sound, and do the data support the conclusions?

Reviewer #1: Yes

Reviewer #2: Yes

3. Has the statistical analysis been performed appropriately and rigorously? 

Reviewer #1: Yes

Reviewer #2: Yes

4. Have the authors made all data underlying the findings in their manuscript fully available?

Reviewer #1: Yes

Reviewer #2: Yes

5. Is the manuscript presented in an intelligible fashion and written in standard English?

Reviewer #1: Yes

Reviewer #2: Yes

6. Review Comments to the Author

Reviewer #1: (No Response)

Reviewer #2: Thank you for your responses. The authors have addressed the concerns listed in the initial review of the paper.

7. PLOS authors have the option to publish the peer review history of their article (what does this mean?). If published, this will include your full peer review and any attached files.

Reviewer #1: No

Reviewer #2: No

---

## [Editor Report · Acceptance letter]

1 Jun 2020

PONE-D-20-05544R1 

Prognostic factors and quality of life after pelvic fractures. The Brabant Injury Outcome Surveillance (BIOS) study. 

Dear Dr. Brouwers:

I am pleased to inform you that your manuscript has been deemed suitable for publication in PLOS ONE. Congratulations! Your manuscript is now with our production department. 

With kind regards,

on behalf of

Dr. Zsolt J. Balogh 

Academic Editor

PLOS ONE